A Python script to design primers for overlap extension PCR to ligate two DNA fragments

Hu Yunzhuo 1 2
Xu Fei 3
Huang Bifang 3
Chen Xuanyang cxy@fafu.edu.cn 1 2
Lin Shiqiang linshiqiang@fafu.edu.cn 1 3
1 Fujian Agriculture and Forestry University, Key Laboratory of Crop Biotechnology , Fuzhou , Fujian , China
2 Fujian Agriculture and Forestry University, College of Agronomy , Fuzhou , Fujian , China
3 Fujian Agriculture and Forestry University, College of Life Science , Fuzhou , Fujian , China
Franco Bernardo
Electronic publication date: 2022 Oct 31
Publication date: 2022
Volume: 10
Electronic Location ID: e14283
Received 2022 Aug 8; Accepted 2022 Sep 30
Copyright: ©2022 Hu et al.
Copyright year: 2022
Copyright holder: Hu et al.
License: This is an open access article distributed under the terms of the Creative Commons Attribution License, which permits unrestricted use, distribution, reproduction and adaptation in any medium and for any purpose provided that it is properly attributed. For attribution, the original author(s), title, publication source (PeerJ) and either DOI or URL of the article must be cited.
License URL: https://creativecommons.org/licenses/by/4.0/

Keywords: DNA joining, Overlap extension PCR, Primer design, Python script

Funding: Project of Fujian Province Department of Science & Technology 2020N5013 This work was supported by the Project of Fujian Province Department of Science & Technology (2020N5013). The funders had no role in study design, data collection and analysis, decision to publish, or preparation of the manuscript.

==============================
Ligating two or more DNA fragments is a regular operation for the subcloning and the engineering of vectors. The overlap extension PCR serves as a straightforward method to solve this issue. However, it takes a relatively long time to design the appropriate overlapping primers and the primers for the full-length sequence, and there has not been a professional offline software for such kind of primer design. Here, we propose a Python script to search, calculate and sort thousands of combinations of primers for users according to the predefined parameters. The results of script running and experimental validation show that this script is capable of generating the optimal pairs of primers based on the proper melting temperatures and lengths of the primers, which facilitates gene modification in research.

Introduction

It is a general experiment to ligate several DNA fragments during the construction of plasmids for exploring gene function. The familiar scenarios include adding specific promoters or terminators to the genes of interest, joining tandem gene duplicates, and fusing reporter genes such as GFP, RFP, or YFP to the target genes (Loehlin & Carroll, 2016; O’Halloran, Uriagereka-Herburger & Bode, 2017). In terms of assembling multiple overlapping DNA sequences, the Gibson method is highly efficient and widely applied in gene synthesis (Gibson et al., 2009). In contrast, in less complicated cases, such as ligation of two DNA fragments, the approach of overlap extension PCR is enough and can be easily carried out.

The method of overlap extension PCR takes the first and second DNA fragments as a whole sequence in the design of overlapping primers, consisting of nucleotides from both fragments. Amplifications are carried out to produce the upstream fragment and the downstream fragment separately with the corresponding 5′ and 3′ primers. Then, the upstream and downstream fragments are used as the template DNA of overlap extension PCR to obtain the full-length chimera fragment that can be applied in the following steps, e.g., ligation, sequencing (Fig. 1).

Figure 1 The procedure of ligating two DNA fragments via overlap extension PCR.

The first fragment is shown with a long blue arrow (sense strand) and a long orange arrow (antisense strand). The second fragment is shown with a long cyan arrow (sense strand) and a long purple arrow (antisense strand). The arrows show the 5′–3′ direction of DNA. The PCR#1 shows the amplification of the upstream fragment with primer gene_5 and primer overlap_3. The PCR#2 shows the amplification of the downstream fragment with primer overlap_5 and primer gene_3. The upstream and downstream fragments bridge within the overlapping region during the overlap PCR of the full-length chimera sequence.

Although this method is simple, low-cost and highly efficient, the design of the overlapping primers and the primers for the full-length sequence is time-consuming and tedious. To make the two pairs of primers work well in three PCR reactions, the annealing temperatures of all the primers should be carefully considered. Though there has been an online tool for designing such primers (O’Halloran, Uriagereka-Herburger & Bode, 2017), there is no convenient software that can do the primer design offline by far. In this study, we put forth a method to design the two pairs of primers for the overlap extension PCR using Python program. The algorithm is explicit, and the source code is provided with detailed annotation, which is convenient for users to modify based on personal needs. We also performed experiments to verify the script by using the recommended primers to ligate the two example genes. It is our hope that the script will support the gene cloning process.

Material and Methods

Materials

The computer system used in this study is macOS Monterey 12.3.1 or higher (Apple Inc., CA, USA). Python 3.10.4 (http://www.python.org) and Biopython 1.79 are required for running the script ‘Overlap_ligation_primer_v1.py’ (Cock et al., 2009). Users can modify the source code of the script using the IDLE (Integrated Development and Learning Environment) (http://www.python.org). The sequences of the two DNA fragments are stored in fasta files. The first DNA sequence file ECdnaQ.fasta contains the sequence of the dnaQ from Escherichia coli str. K-12 substr. MG1655 (ECdnaQ) with the gene ID 946441, and the second DNA file TBdnaN.fasta contains the sequence of the dnaN from Mycobacterium tuberculosis H37Rv (TBdnaN) with the gene ID 887092. The above script, fasta files, readme file and the zipped output file can be downloaded via the link at https://github.com/shiqiang-lin/PCR-ligate-2_DNAs.

Algorithm

In this script, the following rules are applied in the design of the overlapping primers and the primers for the full-length sequence. (1) The two overlapping primers (Fig. 1), each containing the nucleotides from both the two fragments, are partially reverse complemented, with 3′ overhangs (Fig. 2). (2) The Tm values of the four primers should be similar, which facilitates the PCR reactions. (3) The Tm value of the overlapping region between the two primers (Fig. 2) should be moderate. In our program, we set the favorite Tm value 63 °C. However, users can modify this value by changing the variable in the line 90 of the source code (which is ‘Tm_favorite = 63’). (4) The combinations of the four primers are sorted according to the K value, which is defined with the formulae below.

Figure 2 A schematic illustration of the overlapping primers.

The arrows show the 5′–3′ direction of DNA. The primers are partially complemented with 3′ overhangs. The upper arrow indicates the forward overlapping primer, consisting of the nucleotides (blue) from the 3′ end of the sense strand of the first DNA fragment and the nucleotides (cyan) from the 5′ end of the sense strand of the second DNA fragment. The lower arrow indicates the reverse overlapping primer, consisting of the nucleotides (purple) from the 3′ end of the antisense strand of the second DNA fragment and the nucleotides (orange) from the 5′ end of the antisense strand of the first DNA fragment. The length of each part is labeled, with i, j, k, l all ranging from 0 to 3 bps (3 bps included). The variations of i, j, k, l enable us to walk through all possible combinations for the overlapping primers. The vertical gray dashes show the overlapping region between the two primers, which is also used for bridging during overlap extension PCR to acquire the full-length chimera DNA sequence.

K=100×Tmprimer5−Tmoverlap3+Tmoverlap5−Tmgene3+Tmprimer5−Tmprimer3+Tmpimer5+Tmprimer32−Tmoverlap−3+Tmoverlap−Tmfavorite+Tmprimer5+Tmprimer32−Tmfavorite

The ‘Tm’ represents the value of melting temperature of primer. The ‘primer5’, ‘overlap3’ are the primers for the first DNA fragment to be joined and ‘overlap5’, ‘primer3’ are the primers for the second DNA fragment to be joined. The ‘overlap’ is the overlapping region between the two primers. The ‘Tmfavorite’ is the set optimal Tm value of the overlapping region between the two primers (Fig. 2). A lower K value of the combination indicates that the difference in melting temperature among the four primers is smaller.

Running method

The script is run with the following steps.

(1) Make a new directory first, copy the script, ECdnaQ.fasta and TBdnaN.fasta to the directory.

(2) Open the Terminal and change the current working directory to the path of the new directory, and run the following command.

python3.10 Overlap_ligation_primer_v1.py ECdnaQ.fasta TBdnaN.fasta

(3) Once the script finishes the task, a new file named ‘ECdnaQ_TBdnaN_ligation_primers.txt’ is generated, which lists the resulting primers (Fig. 3). The process takes less than one minute for a MacBook computer.

Figure 3 File contents (partial) of the ECdnaQ_TBdnaN_ligation_primers.txt.

Experimental validation

The four primers of the first combination in Fig. 3 were synthesized, i.e., gene_5 (ATGAGCACTGCAATTACACGCCAGATCGTT), gene_3 (TCAGCCCGGCAACCGAACCGG), overlap_5 (GGCGAGCATAAATGGACGCGGCTACG) and overlap_3 (GCCGCGTCCATTTATGCTCGCCAGAG). The gene_5 and overlap_3 were used to amplify the ECdnaQ gene from pETDuet-1-EcdnaQ (Supplemental Information S1) with Phusion® High-Fidelity PCR Master Mix with HF Buffer (Catalog Number M0531S; New England Biolabs, Ipswich, MA, USA) (Lin, Bi & Zhang, 2011). The PCR reaction system (50 uL) contained 20 uL ddw, 2 uL 10 uM gene_5, 2 uL 10 uM overlap_3, 1 uL template plasmid (14.1 ng/uL), and 25 uL 2XPhusion HF PCR Master Mix. The PCR program was 98 °C for 30s; 30 cycles of 95 °C for 30s, 60 °C for 30s, 72 °C for 14s; 72 °C for 5min; 16 °C forever. The overlap_5 and gene_3 were used to amplify the TBdnaN gene from pET28a-TBdnaN (Supplemental Information S2) with Phusion® High-Fidelity PCR Master Mix with HF Buffer (Catalog Number M0531S; New England Biolabs, Ipswich, MA, USA) (Gui et al., 2011). The PCR reaction system (50 uL) contained 20 uL ddw, 2 uL 10 uM overlap_5, 2 uL 10 uM gene_3, 1 uL template plasmid (15.9 ng/uL) and 25 uL 2XPhusion HF PCR Master Mix. The PCR program was 98 °C for 30s; 30 cycles of 95 °C for 30s, 61 °C for 30s, 72 °C for 24s; 72 °C for 5min; 16 °C forever.

The ECdnaQ and TBdnaN were then gel recycled and their concentrations were measured with NanoDrop (Thermo Fisher Scientific Inc., Denver, CO, USA). The concentration of ECdnaQ was 294.4 ng/µL and that of TBdnaN was 245.5 ng/µL. The two genes were mixed with equal molar ratio, according to gene length and concentration and neglecting the GC difference. TE buffer was used to dilute the mixed gene to a concentration of 24 ng/µL, which was then used as the template DNA for producing the full-length fragment. The PCR reaction system (50 uL) contained 20 uL ddw, 2 uL 10 uM gene_5, 2 uL 10 uM gene_3, 1 uL mixed genes and 25 uL 2XPhusion HF PCR Master Mix. The PCR program was 98 °C for 30s; 30 cycles of 95 °C for 30s, 60 °C for 30s, 72 °C for 40s; 72 °C for 5min; 16 °C forever. The full-length fragment was gel recycled and ligated to T vector with pEASY®-Blunt Zero Cloning Kit (Catalog Number CB501-01, TransGen Biotech Co., Ltd., Haidian, Beijing, China). The sequence of the ligated fragment was confirmed by Sanger sequencing (Supplemental Information S3).

Moreover, we obtained the full-length fragment based on overlap extension (without adding primers gene_5/gene_3). The PCR reaction system (50 uL) contained 24 uL ddw, 1 ul equal molar mixed genes (262 ng/uL) and 25 uL 2X Phusion HF PCR Master Mix. The PCR program was 98 °C for 30s; 30 cycles of 95 °C for 30s, 60 °C for 30s, 72 °C for 40s; 72 °C for 5min; 16 °C forever. The target band was gel recycled and ligated to T vector with pEASY®-Blunt Zero Cloning Kit (Catalog Number CB501-01, TransGen Biotech Co., Ltd., Haidian, Beijing, China). The sequence of the ligated fragment was confirmed by Sanger sequencing (Supplemental Information S4).

Results

Script running

As shown in Fig. 2, the overlapping primers vary with the values of i, j, k and l. Since i and j both can be one of the four numbers (0, 1, 2, and 3), there are 4*4 = 16 possibilities for the overlapping forward primer. Similarly, there are also 16 possibilities for the overlapping reverse primer. Meanwhile, the search interval of the primer for the full-length sequence is from 14 to 30, therefore, there are 17 possibilities for each primer of full-length sequence. In total, the number of combinations for the overlapping primers and the primers of the full-length sequence is 73,984, equaling 16*16*17*17. Although the number is remarkable, the program can generate all of these combinations within one minute.

An example of the output file ‘ECdnaQ_TBdnaN_ligation_primers.txt’ is shown in Fig. 3. There are seven columns in the output file: the left five columns are the name, sequence, length, GC percentage, Tm value of each primer, and the right two columns are Overlap_Tm and K value. The Overlap_Tm indicates the Tm value of the overlapping region of the overlapping primers in the combination. The K value represents the difference of Tm value among these four primers. Lower K value suggests the combination of the four primers is theoretically preferable. In the example, the K value of the first combination is 583, which is the lowest among the combinations. Although the GC percentage of the gene_3 primer is a little high, this combination is still the best choice since the GC percentage in the 3′ end of the second DNA fragment is intrinsically high.

Experimental validation

The experimental validation process was divided into three steps, i.e., obtaining gel recycled ECdnaQ and TBdnaN fragments, amplifying the full-length fragment via overlap PCR, ligating the full-length fragment to T vector for sequencing confirmation. In Fig. 4, lane 1 and lane 2 showed that the primers picked up by the script (gene_5 and overlap_3 for ECdnaQ, overlap_5 and gene_3 for TBdnaN) could amplify the target genes, which were then gel recycled (lane 3 and lane 4) for template DNA to produce the full-length fragment. Since there were two methods to obtain the full-length fragment, i.e., with primers gene_5/gene_3, without primers gene_5/gene_3, we performed PCR reaction for the two situations respectively. Lane 5 and lane 6 showed that both methods were capable of yielding full-length fragments. The target bands were gel recycled and ligated to T vectors respectively. The sequencing results showed that the sequences of the full-length fragment were correct (Supplemental Information S3–S4). These results demonstrated that the primers recommended by the script could be used to ligate the ECdnaQ and TBdnaN with overlap PCR.

Figure 4 Experimental verification of primers selected by the script.

Lane M, DNA marker; lane 1, PCR amplification of ECdnaQ with primers gene_5/overlap_3 from pETDuet-1-ECdnaQ; lane 2, PCR amplification of TBdnaN with primers overlap_5 and gene_3 from pET28a-TBdnaN; lane 3, gel recycling of the ECdnaQ band from lane 2; lane 4, gel recycling of the TBdnaN band from lane3; lane 5, PCR amplification of full-length fragment with primers gene_5/gene_3 from mixed template DNA of ECdnaQ and TBdnaN; lane 6, producing full-length fragment via overlap extension without primers gene_5/gene_3; lane 7, negative control, the same as lane 5 but using ddw as template. The lowest arrow in the right side indicates the position of ECdnaQ. The middle arrow in the right side indicates the position of TBdnaN. The highest arrow in the right side indicates the position of full-length chimera fragment.

Discussion

In this study, the design of the overlapping primers and the primers of the full-length sequence for combining the two DNA fragments using overlap extension PCR is accomplished by a Python program. As the source code is provided, it is convenient for users to adjust the relevant parameters, such as the searching space for each primer, the optimal Tm value for the Tm_overlap, and so on. The script can also be used to design the deletion primers for overlap extension PCR, in which the input sequences are the two fragments from the target gene with the deleted region removed (Forloni, Liu & Wajapeyee, 2018; Lee et al., 2010). However, both fragments should be no less than 200 bps to ensure an efficient gel extraction.

For amplifying full-length fragment with mixed ECdnaQ and TBdnaN as template DNA, two approaches were used, which were with primers gene_5/gene_3 and without primers gene_5/gene_3. In terms of approach without primers gene_5/gene_3, the double strands of ECdnaQ and TBdnaN could form two kinds of overlaps during the annealing stage, i.e., overlap with two 3′ ends (equivalent to primers) that could elongate by incorporating dNTPs to form the full-length fragment and overlap with two 5′ ends that could not grow in length via DNA polymerase. Due to the addition of plenty of mixed ECdnaQ and TBdnaN, we could obtain enough full-length fragment for T vector ligation.

For the approach with primers gene_5/gene_3, the reaction was more complex. What happened to the reaction without primers gene_5/gene_3 could happen to that with primers gene_5/gene_3. Besides, the overlap with two 5′ ends could be used as template DNA to produce the strands that could form overlap with two 3′ ends in the next round of annealing, which then yielded the double-stranded full-length fragment. These full-length fragments then could be used as template DNA for exponential amplification with primers gene_5/gene_3.

The approach without primers gene_5/gene_3 was simpler as there was no exponential amplification. Therefore, lots of template DNA was needed to guarantee enough full-length fragment for the next step of experiment. A possible benefit of this approach is that the possibility of nucleotide mismatch might be reduced as the reaction was simpler. In comparison, the advantage of the approach with primers gene_5/gene_3 was that only a small amount of mixed template DNA of ECdnaQ and TBdnaN was needed. Due to the complexity of the reaction, this approach might have a slightly lower fidelity than the approach without primers gene_5/gene_3. However, the high-fidelity DNA polymerase used in this experiment ensured that the correct full-length fragments were obtained.

Currently, to the best of our knowledge, there is no other program like ours that is standalone, open-source and can automatically design thousands of primer combinations for performing overlap extension PCR to ligate two DNA fragments. The script is free, annotated in detail so that users can read the code easily and know the algorithm instead of dealing with a black box, which is important for tailoring the script according to the specific, if any, experimental needs. Also, the script exhibits the results in a friendly style for users to pick up the preferred tetrad of primers, which may save time for users and avoid mistakes that might lead to undesired mutation. Furthermore, we provided two approaches for yielding the full-length fragment and showed that both worked, demonstrating the flexibility of our script in the application.

Conclusion

To sum up, we offer a simple and efficient script to design the four primers to ligate the two DNA fragments and the script is verified by experiments. It is our hope that the script may help for doing gene cloning in the lab.

Supplemental Information

Supplemental Information 1 Sequencing result: pET28a-dnaQ

Click here for additional data file.

Supplemental Information 2 Sequencing result: pET28a-dnaN

Click here for additional data file.

Supplemental Information 3 Sequencing result: full-length with primers

Click here for additional data file.

Supplemental Information 4 Sequencing result: full-length without primers

Click here for additional data file.

Additional Information and Declarations

Competing Interests

Author Contributions

Data Availability

The authors declare there are no competing interests.

Yunzhuo Hu performed the experiments, analyzed the data, prepared figures and/or tables, and approved the final draft.

Fei Xu performed the experiments, analyzed the data, prepared figures and/or tables, and approved the final draft.

Bifang Huang analyzed the data, prepared figures and/or tables, and approved the final draft.

Xuanyang Chen conceived and designed the experiments, authored or reviewed drafts of the article, and approved the final draft.

Shiqiang Lin conceived and designed the experiments, authored or reviewed drafts of the article, and approved the final draft.

The following information was supplied regarding data availability:

The sequencing results of genes used in this study are available in the Supplementary Files.

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
