# Peer review of "A Python script to design primers for overlap extension PCR to ligate two DNA fragments"

_PeerJ, doi:10.7717/peerj.14283_

## Round 0.1 · original submission · Major Revisions

Dear authors, thank you so much for submitting your work to PeerJ.

I kindly request to address the issues found by the two experts on your manuscript. The major issue was found by Reviewer 1 and myself, the software has some issue while running. I also kindly request to provide a more detailed troubleshooting guide as a supplementary material. Please revise your software and manuscript to reevaluate the manuscript. Please include a rebuttal letter addressing all the concerns point by point.

Once again, thank you for choosing PeerJ.

Best regards,
Bernardo

Reviewer 1 ·

Basic reporting

no comment

Experimental design

Concerns with the specificity of Fig. 6. There are many background bands and non-specific binding. Also, it is unclear which is the fusion product - this is the important product to display on the gel.

Validity of the findings

non comment

Additional comments

Authors need to provide a more detailed guide for using the script and installing required dependencies. When I ran the script, it worked for the most part, however, the K values were empty in the result file. Also, obtained the following error:

SyntaxWarning: "is" with a literal. Did you mean "=="?
if i.isalpha() or i is "#":

By having the code available on Github as a repo, users will be able to open tickets with an issues so that the code can be improved and bugs identified. This is really my biggest critique of the software. Also, in the results file the overlap primer does not appear to be supplied. This should be made clear. Finally, it would be better if the sequences could be supplied in FASTA format. This is the standard format for DNA sequences and should be easy for the authors to parse. Currently the sequences are in raw format with a # sign separating the sequence which is very unusual.

Reviewer 2 ·

Basic reporting

no comment

Experimental design

no comment

Validity of the findings

no comment

Additional comments

1. The python script reported by this manuscript is useful for overlap PCR design that it is open source so that could be tailored to the users needs. It would be more helpful if the author could add some trouble shooting manual or indicating which of those parameters could adjusted and how they might influence the result.
2. Figure1: The first and the second gene fragment is better to be drawn in different lines rather than next to each other or it will look like they are connected already. It would be helpful to label each step beside the arrow indicating the process like PCR#1, PCR#2, overlap PCR and would also be helpful to name each primer the same as indicated in the function in the text.

---

## Round 0.2 · Minor Revisions

Dear authors,

Please complete the last corrections of your manuscript according to reviewer 1 latest comments. Thank you so much!

Best regards,
Bernardo

Reviewer 1 ·

Basic reporting

no comment

Experimental design

no comment

Validity of the findings

no comment

Additional comments

Please include details on the GitHub Repo about how do download Biopython e.g. link to https://biopython.org/wiki/Download or type pip install biopython

The authors should include Overlap_mutagenic_primer_v1.py in the repo - or at the very least link to where to find it

Reviewer 2 ·

Basic reporting

no comment

Experimental design

no comment

Validity of the findings

no comment

Additional comments

The authors answered all the questions, improved the figures and made detailed user guide available to the public through Github. The algorithm is well explained and the code is well annotated. This python script will have general interest to the scientific community. Recommend to accept as is.

---

## Round 0.3 · accepted · Accept

Dear authors,

Thank you for submitting your work to PeerJ. I am happy to announce to you that with the latest corrections your manuscript is now ready for publication.

Thank you for correcting the last minor issues found. I also thank the two reviewers that provided positive feedback to this manuscript. I encourage authors to keep updated the software and fix any future bugs that may be detected.

Once again, thank you, and congratulations!

Best regards,

Bernardo